# Are Leading Risk Factors for Cancer and Mental Disorders Multimorbidity Shared by These Two Individual Conditions in Community-Dwelling Middle-Aged Adults?

**DOI:** 10.3390/cancers12061700

**Published:** 2020-06-26

**Authors:** Xianwen Shang, Allison M. Hodge, Wei Peng, Mingguang He, Lei Zhang

**Affiliations:** 1Centre for Eye Research Australia, Royal Victorian Eye and Ear Hospital, Melbourne, VIC 3002, Australia; xianwen.shang@unimelb.edu.au; 2School of Behavioural and Health Sciences, Australian Catholic University, Melbourne, VIC 3002, Australia; 3Department of Medicine (Royal Melbourne Hospital), University of Melbourne, Melbourne, VIC 3050, Australia; 4Cancer Epidemiology Division, Cancer Council Victoria, Melbourne, VIC 3004, Australia; Allison.Hodge@cancervic.org.au; 5Centre for Epidemiology and Biostatistics, Melbourne School of Population and Global Health, University of Melbourne, Melbourne, VIC 3010, Australia; 6Research Centre for Data Analytics and Cognition, La Trobe University, Melbourne, VIC 3083, Australia; weipengtiger@yahoo.com.au; 7Ophthalmology, Department of Surgery, University of Melbourne, Melbourne, VIC 3002, Australia; 8State Key Laboratory of Ophthalmology, National Clinical Research Center, Zhongshan Ophthalmic Center, Sun Yat-sen University, Guangzhou 510060, China; 9China-Australia Joint Research Center for Infectious Diseases, School of Public Health, Xi’an Jiaotong University Health Science Centre, Xi’an 760061, Shaanxi, China; 10Melbourne Sexual Health Centre, Alfred Health, Melbourne, VIC 3053, Australia; 11Central Clinical School, Faculty of Medicine, Monash University, Melbourne, VIC 3004, Australia; 12School of Public Health and Preventive Medicine, Faculty of Medicine, Monash University, Melbourne, VIC 3004, Australia

**Keywords:** multimorbidity, cancer, mental disorders, shared risk factors, leading risk factors, machine learning

## Abstract

Data on the leading shared risk factors of cancer and mental disorders are limited. We included 98,958 participants (54.8% women) aged 45–64 years from the 45 and Up Study who were free of cancer, depression, and anxiety at baseline (2006–2009). The incidence of cancer, mental disorders, and multimorbidity (concurrent cancer and mental disorders) was identified using claim databases during follow-up until 31 December 2016. During a nine-year follow-up, the cumulative incidence of cancer, mental disorders, and multimorbidity was 8.8%, 17.4%, and 2.2%, respectively. Participants with cancer were 3.41 times more likely to develop mental disorders, while individuals with mental disorders were 3.06 times more likely to develop cancer than people without these conditions. The shared risk factors for cancer and mental disorders were older age, female gender, smoking, psychological distress, low fruit intake, poor/fair self-rated health, hypertension, arthritis, asthma, and diabetes. Low education, low income, overweight/obesity, and family history of depression were risk factors for mental disorders and multimorbidity but not cancer. In conclusion, smoking, low fruit intake, and obesity are key modifiable factors for the prevention of cancer and mental disorders. Individuals with poor/fair self-rated health, high psychological distress, asthma, hypertension, arthritis, or diabetes should be targeted for the prevention and screening of cancer and mental disorders.

## 1. Introduction

Global cancer cases increased from 13.4 million in 2006 to 17.2 million in 2016 [1], and this number is expected to grow rapidly in the next decade due to population growth and aging [1]. In 2015, cancer, which accounted for over 8.7 million deaths, was the second leading cause of global mortality [2]. In Australia, in 2013, cancer of all types surpassed cardiovascular disease as the leading cause of death [3]. Around one-third of individuals (29.2%) surveyed in high-income countries over the last two decades were affected by a common mental disorder at some time during their lifetimes [4]. It was estimated that 14.3% of deaths [5], 32.4% of years lived with disability, and 13.0% of disability-adjusted life-years worldwide are attributable to mental disorders [6]. Mental disorders commonly cluster with cancer [7,8,9], and the presence of mental disorders is associated with an increased risk of cancer mortality [10,11]. The concurrence of mental disorders and cancer (multimorbidity) imposes a tremendous burden on the healthcare system.

The common coexistence of cancer and mental disorders suggests there may be shared risk factors for these. Socioeconomic status, medical history, family history of conditions, and behavioral and psychological factors were linked to cancer [12] and mental disorders separately in previous studies [13,14,15]. However, the leading predictors for the multimorbidity were not examined, and whether these predictors are shared by cancer and mental disorders is not known. Determining shared risk factors for mental disorders and cancer would help prevent and manage multimorbidity and minimize deaths caused by these concurrent conditions.

We evaluated the importance of 48 potential predictors on the development of multimorbidity using machine learning methods. We then analyzed whether the selected leading predictors for multimorbidity were independently associated with incident multimorbidity, as well as cancer and mental disorders.

## 2. Results

### 2.1. Participant Characteristics

As shown in Table 1, 98,958 participants (54.8% women) with a mean follow-up of 8.9 ± 0.9 (7.0–11.5) years were included in the final analysis. Individuals aged 45–54 years were more likely to have higher income and education, smoke, and consume less vegetables, fruit, and red meat and more chicken compared with those aged 55–64 years (all *p* < 0.0001).

### 2.2. Incidence of Cancer, Mental Disorders, and Multimorbidity

During follow-up, the cumulative incidence of multimorbidity was 2.2% with women (2.6%) having a higher risk (hazard ratio (HR) (95% confidence intervals (CI)): 1.58 (1.42–1.76)) than men (1.7%). Women had a higher incidence of both cancer (9.3% vs. 8.3%) and mental disorders (20.6% vs. 13.7%) than men. In the multivariable analysis, individuals aged 55–64 years had a higher risk of incident cancer, mental disorders, and multimorbidity in both men and women than those aged 45–54 (Figure 1A).

In multivariable analysis, participants with cancer as the primary condition were 3.41 (3.16–3.67) times more likely to develop mental disorders compared with the total population who developed mental disorders as the primary condition (39.6% vs. 11.7%). The corresponding HRs (95% CIs) were 4.03 (3.54–4.57) for men and 3.08 (2.81–3.38) for women, and this association was more likely to be manifested in men (*p* for interaction <0.0001, Figure 1B).

Individuals with mental disorders as a primary condition had a higher risk (HR (95% CI): 3.06 (2.78–3.36)) of developing cancer as the secondary condition compared with the general population (17.9% vs. 6.2%) who developed cancer as the primary condition (Figure 1C).

### 2.3. Leading Predictors for Multimorbidity

Random forest had a higher performance compared with the other three machine learning models (Appendix A). The common leading predictors for multimorbidity from different methods were older age, female gender, smoking, high chicken intake, asthma, psychological distress, arthritis, hypertension, family history of cancer, diabetes, and poor/fair self-rated quality of life (Appendix A).

### 2.4. Hazard Ratios for Cancer, Mental Disorders, and Multimorbidity Associated with Potential Predictors

Individuals who self-reported poor/fair overall health had a higher risk of developing cancer (HR (95% CI): 1.37 (1.24–1.52), mental disorders (1.73 (1.61–1.85)), and multimorbidity (1.95 (1.60–2.37)) than those with excellent self-rated overall health. Psychological distress was associated with a higher risk of incident cancer, mental disorders, and multimorbidity. In multivariable analysis, low income and low education were risk factors for incident mental disorders and multimorbidity but not cancer. Individuals with hypertension were more likely to develop cancer (HR (95% CI): 1.13 (1.07–1.18)), mental disorders (1.15 (1.11–1.19)), and multimorbidity (1.25 (1.13–1.38)) than those without hypertension. Asthma, arthritis, and diabetes at baseline were each associated with a higher incidence of cancer, mental disorders, and multimorbidity. Family history of depression was associated with a higher incidence of mental disorders and multimorbidity but not cancer. Family history of cancer was a risk factor for incident cancer and multimorbidity but not mental disorders (Figure 2).

Similar results were observed in both genders (Appendix A).

Individuals who smoked at baseline had a higher risk of incident cancer (HR (95% CI): 1.29 (1.20-1.40)), mental disorders (1.58 (1.50-1.66)), and multimorbidity (1.68 (1.46-1.93)) than those who never smoked. High fruit intake was associated with a lower risk of developing mental disorders and cancer. Obesity at baseline was associated with a higher risk of incident mental disorders and multimorbidity but not cancer. Normal sleep duration (7–9 h/night), high vegetable intake, and high levels of physical activity were associated with lower risks of incident mental disorders but not cancer (Figure 3).

These associations were significant in both men and women (Appendix A).

The risk factors could be categorized into four groups: (1) shared risk factors for cancer, mental disorders, and multimorbidity; (2) risk factors for mental disorders and multimorbidity; (3) risk factor for cancer and multimorbidity; (4) risk factors for mental disorders only (Appendix A).

### 2.5. Sensitivity Analysis

Risk factors for cancer and mental disorders multimorbidity were positively associated with the incidence of cancer and depression multimorbidity (Appendix A). Participants with cancer at baseline were more likely to develop mental disorders during follow-up (HR (95% CI): 1.14 (1.10–1.18)) compared those without cancer at baseline (Appendix A). The cumulative incidence of cancer among 26,853 individuals with mental disorders at baseline (9.1%) was not higher (*p* value = 0.32) than those without mental disorders at baseline (8.8%) (Appendix A).

## 3. Discussion

In this large-scale cohort study, we found that both baseline and new-onset cancer were associated with an increased risk of subsequent incident mental disorders, while new-onset but not baseline mental disorders were associated with an increased risk of subsequent incident cancer. The shared risk factors for cancer and mental disorders were older age, female gender, smoking, low fruit intake, poor/fair self-rated health, psychological distress, hypertension, arthritis, asthma, and diabetes. Low education, low income, overweight/obesity, and family history of depression were risk factors for incident mental disorders and multimorbidity, while family history of cancer was a risk factor for incident cancer and multimorbidity. Short/long sleep duration, low vegetable intake, high chicken intake, and physical inactivity were risk factors for mental disorders only.

To our knowledge, this is the first study to explore the leading risk factors for cancer and mental disorders multimorbidity using machine learning methods and examine whether the leading risk factors for multimorbidity were shared by cancer and mental disorders. We identified the shared risk factors that may have the potential to simultaneously impact on the development of cancer and mental disorders, and these factors are priorities for intervention for the prevention of cancer, mental disorders, and multimorbidity.

Our study is consistent with previous studies showing that women had a higher incidence of mental disorders than men [4]. However, unlike some studies [16], we found women had a higher incidence of cancer than men. This may be partly attributed to the age range in our study. Australian statistics for people aged 45–54 years show that women have a higher prevalence of cancer than men, with a similar mortality rate [17]. Our study agrees with previous studies showing that age was strongly associated with the incidence of cancer [1] and weakly associated with incident mental disorders [18].

We observed that individuals with new-onset cancer were at a higher risk of subsequent incident mental disorders than the general population and vice versa in both men and women. Although this interplay was not reported previously, our findings are supported by previous studies demonstrating that existing cancer was associated with a higher risk of mental disorders or vice versa [19,20]. We further found that the positive association between existing cancer and incidence of mental disorders was more evident in men than women, which was not reported in previous studies. A cohort study showed that men had higher HRs for mental disorders associated with cancer three to 10 years after diagnosis compared with women [21]. This is consistent with our findings, although Lu et al. [21] did not exclude participants with self-reported history of mental disorders at baseline in the analysis [21]. We found cancer at baseline was associated with an increased risk of incident mental disorders over 11 years, although the HR (95% CI), 1.14 (1.10–1.18), was lower compared with 3.41 (3.16–3.67) for incident mental disorders within seven years following the onset of cancer. Consistent with our observations, a previous study reported that the magnitude of the positive association between existing cancer and incident mental disorders decreased with increased time after cancer diagnosis [21]. A medical history or family history of mental disorders was not associated with incident cancer in our study. The positive association between depression and the risk of cancer was less evident in studies with a follow-up greater than 10 years [20]. The association between cancer and mental disorders is significant, especially in the short-term, suggesting the importance of incorporating psychological care into cancer treatment and cancer screening in people with mental disorders promptly following diagnosis [22].

Smoking was a leading risk factor for multimorbidity, and both former and current smoking was associated with a higher risk of cancer and mental disorders in our study. Smoking was found to be among leading risk factors for cancer death [12]. The positive associations of smoking with mental disorders and many types of cancers, particularly lung and colorectal cancers, are well established [15,23,24]. We also observed there was a stronger positive association of smoking with multimorbidity than with either cancer or mental disorders alone. This needs to be investigated in more prospective studies.

We observed that obesity was associated with a higher risk of some cancers (except breast cancer and melanoma), as well as mental disorders and multimorbidity. It was reported that overweight/obesity was a risk factor for mental disorders [15] and cancers of some but not all sites [25,26]. Despite this, it was estimated that the population-attributable fraction for all cancers in Australia from overweight/obesity was 5% for deaths and 4% for incidence [27]. Previous studies also demonstrated that obesity was not associated with lung cancer and had divergent associations with premenopausal and postmenopausal breast cancers [28]. This partly explains why no significant association between obesity and the incidence of all cancers was observed in our study.

Three or more servings/day of fruit intake was associated with a lower risk of incident cancer and mental disorders in our study. This agrees with previous studies highlighting the importance of fruit intake for the prevention of cancer and mental disorders [12,15]. We found that high vegetable intake was associated with a lower incidence of mental disorders but not cancer. One explanation for this may be that the association is for smoking-related cancers [29]. However, we cannot distinguish those smoking-related cancers in our study. Physical activity was inversely associated with incident mental disorders but not cancer in our study. A recent meta-analysis of 1.44 million adults demonstrated that higher physical activity was associated with lower risks of 10 types of cancer and higher risks of two cancers (melanoma and prostate cancer), while not associated with the incidence of another 14 cancers [30]. The population-attributable fraction for all cancers from physical inactivity was 0.8% for mortality and 1.5% for incidence [27]. The small contributioon of physical inactivity to incident cancer and divergent associations of different cancers with physical inactivity might partly explain the lack of the association between physical activity and the incidence of all cancer in our study. Normal sleep duration was associated with a lower risk of depression compared [15] but its association with cancer risk was inconsistent among previous studies [31]. We did not find a significant association between sleep duration and incident cancer. Smoking and low fruit intake are shared risk factors for cancer and mental disorders. Although normal weight, normal sleep duration, high vegetable intake, and high physical activity were not inversely associated with incident cancer, they may help prevent the development of mental disorders. Therefore, these healthy habits should be also promoted.

We found poor/fair self-rated health, poor/fair self-rated quality of life, and psychological distress were all leading predictors for multimorbidity and were shared risk factors for incident cancer and mental disorders. Previous studies reported a positive association of poor/fair self-rated health with the risk of mental disorders [32,33,34] and cancer [33,35]. The high predictive ability may be due to the fact that self-rated health reflects the perception of biological and psychological status of individuals [36]. In line with our study, a recent pooled analysis of data from 16 prospective studies showed a positive association between psychological distress and incident cancer [37]. These self-rated scores may help identify high-risk populations for intervention considering that they are strong independent predictors for cancer, mental disorders, and multimorbidity.

Previous studies demonstrated that chronic conditions may increase risks of cancer or mental disorders [38,39,40,41,42,43]. However, previous studies could not determine whether these conditions were shared risk factors for cancer and mental disorders or whether they were associated with an increased risk of multimorbidity. For example, hypertension, diabetes, asthma, and arthritis were each demonstrated to be risk factors for cancer in some studies [38,39,40] and mental disorders in other studies [41,42,43]. Our study demonstrates that hypertension, diabetes, asthma, and arthritis were among leading predictors for multimorbidity and were also shared risk factors for cancer and mental disorders. This suggests that the prevention and management of these chronic conditions may be important in prevention of cancer and mental disorders.

The strengths of this study included the large sample size and long-term follow-up of a community-dwelling population. To our knowledge, this is the first study to explore the leading predictors for cancer and mental disorders multimorbidity and examine whether the leading risk factors were shared by cancer and mental disorders.

The present study has several limitations. Firstly, all data regarding exposures were self-reported, which might have resulted in bias. However, the measurement errors would be more likely to bias true associations to the null. It is possible that some important predictors might be missed as the leading risk factors from machine learning and the association of some leading risk factors with the outcomes might be underestimated in the Cox regression models. Furthermore, incident cancer/mental disorders were identified by treatment using procedures or medications, thus possibly resulting in an underestimated incidence. It is possible that some cases of mental disorders might occur ahead of cancer but the treatment temporal order might be reversed. Thirdly, several types of cancer (lung cancer or colorectal cancer) might not be accurately identified using Medicare Benefits Schedule (MBS) and Pharmaceutical Benefits Scheme (PBS), which did not allow us to analyze whether some risk factors were shared by site specific cancers and mental disorders. The low incidence of multimorbidity (the concurrence of mental disorders and individual types of cancer, including melanoma, prostate cancer, breast cancer, or colorectal cancer) also limited our analysis regarding the shared risk factors. Finally, the participation rate of our study is relatively low, but it is comparable to other cohort studies of this kind [44,45]. Participants in the present study were, on average, healthier than the general population in New South Wales. Similar associations between exposures and health outcomes in this cohort study compared with a population-representative study were reported [46].

## 4. Materials and Methods

### 4.1. Participants

The 45 and Up Study is a prospective study of 266,896 participants aged 45 years and over from New South Wales (NSW) [47]. Participants were randomly sampled from the general population through the Department of Human Services (formerly Medicare Australia) enrolment database and an 18% response rate was achieved, corresponding to 11% of the entire NSW population in the target age group [46]. Baseline data were collected between 2006 and 2009. These data were linked to the Medicare Benefits Schedule (MBS) and Pharmaceutical Benefits Scheme (PBS) data (1 July 2004–31 December 2016) by the Sax Institute using a unique identifier provided by the Department of Human Services. The 45 and Up study has ethical approval from the UNSW Human Research Ethics Committee (HREC 05035). The study protocol was approved by the Royal Victorian Eye and Ear Hospital Human Research Ethics Committee (17/1330HS). Participants provided consent to follow-up and link their data to routine health datasets.

This analysis excluded participants with cancer (excluding non-melanoma skin cancer), heart disease, stroke, depression, anxiety, dementia, and Parkinson’s disease based on self-reported history of previous diagnosis, MBS or PBS claims, those with Department of Veterans′ Affairs cards, those aged 65 years or older, or those who needed help with daily tasks because of long-term illness disability (Appendix A) at baseline. We excluded people with severe diseases, those aged ≥65 years, or those with long-term illness disability, because they were at high risk of mortality, and these conditions were highly correlated with mental disorders. We excluded those with the Department of Veterans′ Affairs cards because information on these people is not included in PBS or MBS.

### 4.2. Independent Variables

An administrative questionnaire was used to collect data. Demographic information including age, gender, ethnicity, income, education, marital status, working status, number of children, and health insurance was self-reported. Behavioral factors including dietary intakes, smoking, alcohol consumption, physical activity, sleep and sitting time were assessed based on structured questions. Body mass index (BMI) was calculated using self-reported height and weight.

Psychological distress, quality of life, overall health, socioeconomic status, and geographic remoteness were assessed individually using indices. Medical history and family history of heart disease, stroke, hypertension, diabetes, cancer, Alzheimer’s disease, Parkinson’s disease, depression, arthritis, and hip fracture were self-reported. The classification of each independent variable is detailed in Text S1. All potential predictors included in the prediction model were only measured at baseline.

### 4.3. Outcome Variables

The incidence of cancer and mental disorders (depression/anxiety) during follow-up was determined by medication and medical services claimed by study participants via PBS or MBS (codes are in Appendix A). Multimorbidity was defined as the concurrence of cancer and mental disorders during follow-up. We regarded cancer and mental disorders as primary or secondary conditions according to their temporal onset order.

### 4.4. Statistical Analysis

Descriptive data were summarized as frequency (percentage), and a chi-square test was used to evaluate the difference of baseline characteristics between age and gender groups. Cox proportional regression models were used to examine whether the incidence of cancer and mental disorders multimorbidity differed by gender and by age groups and to assess the interaction between the onsets of cancer and mental disorders (details in Appendix A).

Given the advantage in prediction performance [48], machine learning was used to identify the leading predictors of incident multimorbidity. Random forest was used to assess the importance of potential predictors for multimorbidity (details inAppendix A). Leading predictors were obtained according to their contribution derived from machine learning with best prediction performance (Appendix A). All available potential predictors in the 45 and Up Study that were known to be associated with cancer or mental disorders were featured in the prediction models.

The follow-up period was calculated from the recruit date in 2006–2009 to the date of onset of cancer, mental disorders, or multimorbidity, or the end of the follow-up, whichever came first. The associations of these selected leading predictors with the incidence of cancer, mental disorders, and multimorbidity were assessed using Cox regression models. The multivariable analysis was adjusted for age, gender, the country of birth, income, education, health insurance, lifestyle factors, medical history, family history of conditions, BMI, self-rated overall health, and quality of life. This analysis was conducted for men and women separately. Multiple imputations for missing data were conducted for baseline covariates, and we included all variables in the imputation models to create 10 imputed datasets.

As depression was a common and serious mental health problem, a sensitivity analysis was conducted to examine whether risk factors for cancer and mental disorders multimorbidity were associated with incident depression and incident multimorbidity of cancer and depression. We also analyzed whether the presence of mental disorders at baseline was a risk factor for incident cancer and whether baseline cancer was a risk factor for incident mental disorders.

We realized the machine learning modeling using the statistical software R 3.4.1 (R Foundation for Statistical Computing, Vienna, Austria). Other analyses were performed using SAS version 9.4 (SAS Institute Inc., Cary, NC, USA). Associations were considered statistically significant at *p* values < 0.05.

## 5. Conclusions

There was a significant interplay between cancer and mental disorders, especially in the short term. Smoking, low fruit intake, and obesity are key modifiable risk factors for the prevention of cancer and mental disorders. Individuals with poor/fair self-rated health and psychological distress should be targeted for prevention and screening. Good management and treatment of hypertension, asthma, arthritis, and diabetes may help delay or prevent the development of cancer and mental disorders.

## Figures and Tables

**Figure 1 cancers-12-01700-f001:**
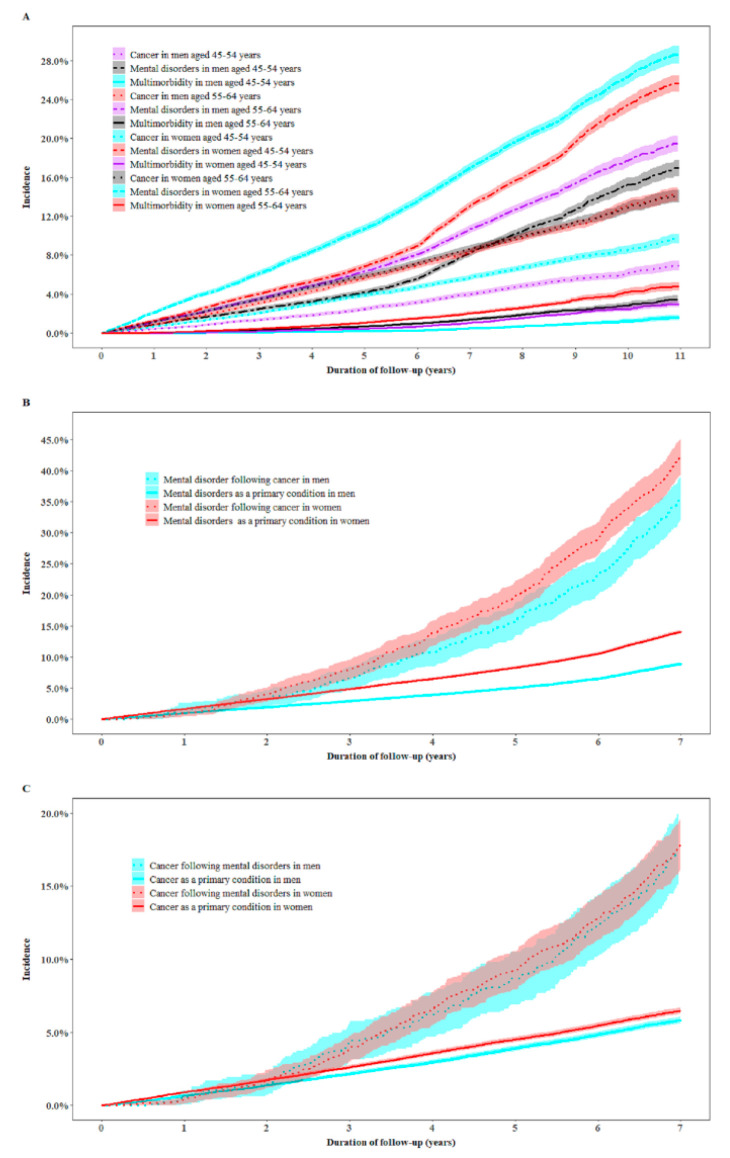
Incidence of cancer, mental disorders, and multimorbidity in middle-aged men and women. (**A**) The cumulative incidence of cancer, mental disorders, and multimorbidity in men and women by age groups. Cox proportional regression models were used to examine whether the incidence of cancer and mental disorders multimorbidity differed by gender and by age groups after adjustment for the country of birth, income, education, health insurance, lifestyle factors, medical history, family history of conditions, BMI (body mass index), self-rated overall health, and quality of life (gender and age were mutually adjusted for). (**B**) The incidence of mental disorders within seven years in individuals with cancer and in the general population by age and gender groups. (**C**) The incidence of cancer within seven years in individuals with mental disorders and in the general population by age and gender groups. To enable comparison, the onset of the primary and secondary condition was restricted to the cases that occurred within the first seven years for this specific analysis, where participants with follow-up time <7 years for the secondary condition were excluded. A cut-off point of seven years was used because the third quartile of the onset time of cancer since baseline was 7.0 years, and that of mental disorders was 7.2 years. Cox proportional regression models were used to assess the interplay between the onsets of cancer and mental disorders after adjustment for the country of birth, income, education, health insurance, lifestyle factors, medical history, family history of conditions, BMI, self-rated overall health, and quality of life.

**Figure 2 cancers-12-01700-f002:**
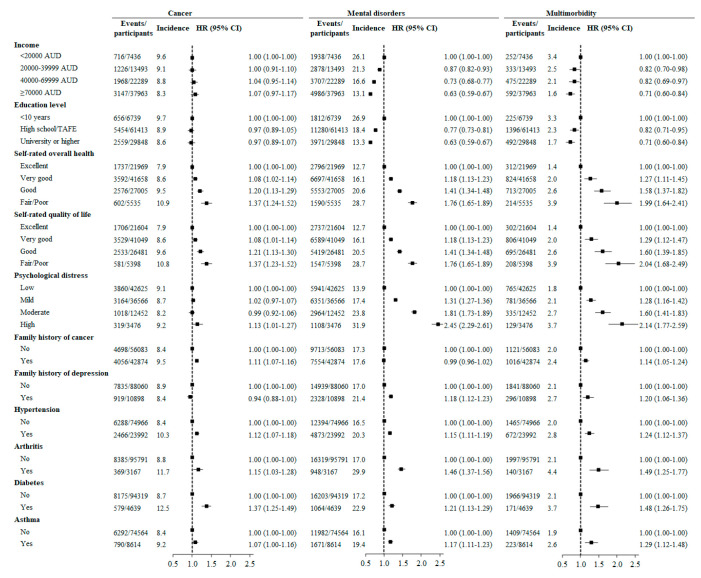
Hazard ratios for incident cancer, mental disorders, and multimorbidity associated with socioeconomic status, self-rated health and psychological distress, and history and family history of chronic conditions after multiple imputation of missing values in covariates. Hazard ratios (HRs) were assessed using Cox regression models adjusted for age, gender, the country of birth, income, education, work status, number of children, BMI, psychological distress, hypertension, dyslipidemia, diabetes, asthma, arthritis, hip replacement, and family history of cancer, depression, heart disease, stroke, diabetes, hypertension, Parkinson’s disease, and dementia. As a minimal difference in HRs (95% confidence intervals (CIs)) before and after multiple imputation of missing values in covariates was observed, we only present the results after multiple imputation in this figure.

**Figure 3 cancers-12-01700-f003:**
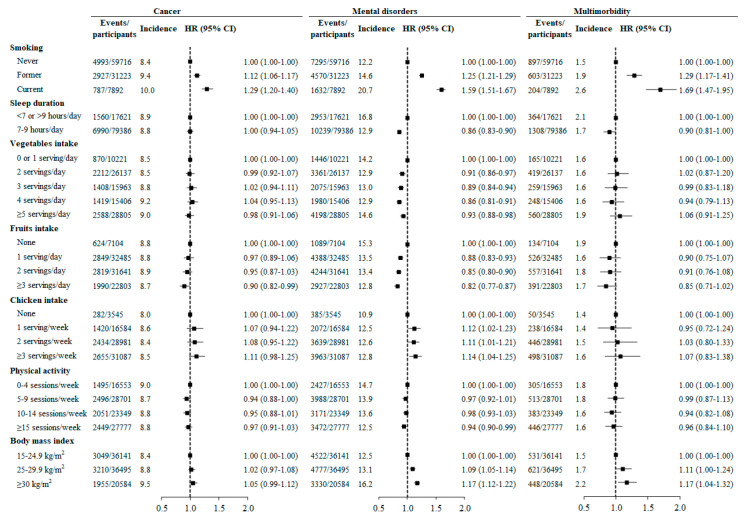
Hazard ratios for incident cancer, mental disorders, and multimorbidity associated with behavioral factors after multiple imputation of missing values in covariates. Hazard ratios were assessed using Cox regression models adjusted for age, gender, the country of birth, income, education, work status, number of children, BMI, psychological distress, hypertension, dyslipidemia, diabetes, asthma, arthritis, hip replacement, and family history of cancer, depression, heart disease, stroke, diabetes, hypertension, Parkinson’s disease, and dementia. As a minimal difference in HRs (95% CIs) before and after multiple imputation of missing values in covariates was observed, we only present the results after multiple imputation in this figure.

**Table 1 cancers-12-01700-t001:** Baseline characteristics of study participants.

Variables	Men		Women	
	45–54 Years	55–64 Years	45–54 Years	55–64 Years
Country of birth	–	–	–	–
Australia	16,659 (75.7)	16,407 (71.7)	21,605 (76.1)	19,042 (74.2)
Others	5260 (23.9)	6352 (27.7)	6692 (23.6)	6471 (25.2)
Missing	93 (0.4)	134 (0.6)	108 (0.4)	135 (0.5)
Household income	–	–	–	–
<20,000 AUD	879 (4.0)	1750 (7.6)	1724 (6.1)	3083 (12.0)
20,000–39,999 AUD	1853 (8.4)	3363 (14.7)	3712 (13.1)	4565 (17.8)
40,000–69,999 AUD	4889 (22.2)	5761 (25.2)	6198 (21.8)	5441 (21.2)
≥70,000 AUD	11,931 (54.2)	8804 (38.5)	11,428 (40.2)	5800 (22.6)
Missing	2460 (11.2)	3215 (14.0)	5343 (18.8)	6759 (26.4)
Education level	–	–	–	–
<10 years	1061 (4.8)	1834 (8.0)	1627 (5.7)	2217 (8.6)
High school/TAFE	13,157 (59.8)	14,059 (61.4)	17,304 (60.9)	16,893 (65.9)
University or higher	7608 (34.6)	6724 (29.4)	9247 (32.6)	6269 (24.4)
Missing	186 (0.8)	276 (1.2)	227 (0.8)	269 (1.0)
Residential rurality ^1^	–	–	–	–
Major cities	12,308 (55.9)	11,971 (52.3)	15,008 (52.8)	12,903 (50.3)
Inner regional	6933 (31.5)	7772 (33.9)	9708 (34.2)	9122 (35.6)
Outer regional	2059 (9.4)	2407 (10.5)	2743 (9.7)	2847 (11.1)
Remote	249 (1.1)	237 (1.0)	363 (1.3)	254 (1.0)
Missing	463 (2.1)	506 (2.2)	583 (2.1)	522 (2.0)
Relative socioeconomic disadvantage ^2^	–	–	–	–
1st quintile	3392 (15.4)	3832 (16.7)	4529 (15.9)	4585 (17.9)
2nd quintile	3957 (18.0)	4434 (19.4)	5429 (19.1)	5132 (20.0)
3rd quintile	4088 (18.6)	4176 (18.2)	5490 (19.3)	4781 (18.6)
4th quintile	4096 (18.6)	4061 (17.7)	5085 (17.9)	4349 (17.0)
5th quintile	5877 (26.7)	5703 (24.9)	7091 (25.0)	6081 (23.7)
Missing	602 (2.7)	687 (3.0)	781 (2.7)	720 (2.8)
Family history of cancer	–	–	–	–
No	12,999 (59.1)	13,069 (57.1)	16,145 (56.8)	13,870 (54.1)
Yes	9013 (40.9)	9824 (42.9)	12,260 (43.2)	11,777 (45.9)
Family history of depression	–	–	–	–
No	19,818 (90.0)	21,039 (91.9)	24,476 (86.2)	22,727 (88.6)
Yes	2194 (10.0)	1854 (8.1)	3929 (13.8)	2921 (11.4)
Body mass index ^3^	–	–	–	–
15–18.4 kg/m^2^	103 (0.5)	108 (0.5)	406 (1.4)	323 (1.3)
18.5–24.9 kg/m^2^	6110 (27.8)	6187 (27.0)	12,755 (44.9)	10,035 (39.1)
25–29.9 kg/m^2^	10,052 (45.7)	10,539 (46.0)	7881 (27.7)	8041 (31.4)
≥30 kg/m^2^	4735 (21.5)	4932 (21.5)	5546 (19.5)	5559 (21.7)
Missing	1012 (4.6)	1127 (4.9)	1817 (6.4)	1690 (6.6)
Smoking	–	–	–	–
Never	12,743 (57.9)	11,940 (52.2)	17,797 (62.7)	17,356 (67.7)
Former	7061 (32.1)	9140 (39.9)	8228 (29.0)	6861 (26.8)
Current	2204 (10.0)	1807 (7.9)	2374 (8.4)	1425 (5.6)
Missing	4 (0.0)	6 (0.0)	6 (0.0)	6 (0.0)
Alcohol consumption	–	–	–	–
None	4231 (19.2)	4117 (18.0)	9538 (33.6)	9055 (35.3)
1–4 sessions/week	4450 (20.2)	4225 (18.5)	7293 (25.7)	6076 (23.7)
5–7 sessions/week	3030 (13.8)	3139 (13.7)	4300 (15.1)	4083 (15.9)
7–14 sessions/week	4537 (20.6)	4978 (21.7)	4959 (17.5)	4428 (17.3)
≥15 sessions/week	5569 (25.3)	6236 (27.2)	2010 (7.1)	1679 (6.5)
Missing	195 (0.9)	198 (0.9)	305 (1.1)	327 (1.3)
Physical activity	–	–	–	–
0–4 sessions/week	4160 (18.9)	3934 (17.2)	4808 (16.9)	3681 (14.4)
5–9 sessions/week	6041 (27.4)	6201 (27.1)	8673 (30.5)	7814 (30.5)
10–14 sessions/week	4690 (21.3)	5115 (22.3)	6976 (24.6)	6588 (25.7)
≥15 sessions/week	6574 (29.9)	6978 (30.5)	7364 (25.9)	6899 (26.9)
Missing	547 (2.5)	665 (2.9)	584 (2.1)	666 (2.6)
Sleep time	–	–	–	–
<7 h	3753 (17.0)	3457 (15.1)	3871 (13.6)	3634 (14.2)
7–9 h	17,419 (79.1)	18,163 (79.3)	23,256 (81.9)	20,641 (80.5)
>9 h	430 (2.0)	843 (3.7)	802 (2.8)	853 (3.3)
Missing	410 (1.9)	430 (1.9)	476 (1.7)	520 (2.0)
Sitting time	–	–	–	–
<8 h	13,526 (61.4)	15,477 (67.6)	19,628 (69.1)	18,924 (73.8)
≥8 h	7611 (34.6)	6308 (27.6)	7327 (25.8)	5077 (19.8)
Missing	875 (4.0)	1108 (4.8)	1450 (5.1)	1647 (6.4)
Chicken intake	–	–	–	–
None	617 (2.8)	892 (3.9)	1021 (3.6)	1017 (4.0)
1 serving per week	3313 (15.1)	4637 (20.3)	4041 (14.2)	4626 (18.0)
2 servings per week	6372 (28.9)	6836 (29.9)	8165 (28.7)	7641 (29.8)
3 or more servings per week	7508 (34.1)	6373 (27.8)	9670 (34.0)	7572 (29.5)
Missing	4202 (19.1)	4155 (18.1)	5508 (19.4)	4792 (18.7)
Fish intake	–	–	–	–
None	2037 (9.3)	1510 (6.6)	2698 (9.5)	1713 (6.7)
1 serving per week	9887 (44.9)	9917 (43.3)	11,772 (41.4)	9415 (36.7)
2 servings per week	5258 (23.9)	6054 (26.4)	7043 (24.8)	7498 (29.2)
3 or more servings per week	3685 (16.7)	4177 (18.2)	5388 (19.0)	5807 (22.6)
Missing	1145 (5.2)	1235 (5.4)	1504 (5.3)	1215 (4.7)
Red meat intake	–	–	–	–
0 or 1 serving per week	2324 (10.6)	2250 (9.8)	4007 (14.1)	3269 (12.7)
2 servings per week	3850 (17.5)	3689 (16.1)	5371 (18.9)	4341 (16.9)
3 or 4 servings per week	7370 (33.5)	7884 (34.4)	9967 (35.1)	9319 (36.3)
5 or more servings per week	4351 (19.8)	5169 (22.6)	3568 (12.6)	4028 (15.7)
Missing	4117 (18.7)	3901 (17.0)	5492 (19.3)	4691 (18.3)
Vegetable intake	–	–	–	–
0 or 1 serving per day	3534 (16.1)	3601 (15.7)	1833 (6.5)	1264 (4.9)
2 servings per day	7635 (34.7)	7113 (31.1)	6528 (23.0)	4884 (19.0)
3 servings per day	3554 (16.1)	3562 (15.6)	4994 (17.6)	3853 (15.0)
4 servings per day	2666 (12.1)	2905 (12.7)	5105 (18.0)	4729 (18.4)
5 or more servings per day	4159 (18.9)	5154 (22.5)	9251 (32.6)	10,298 (40.2)
Missing	464 (2.1)	558 (2.4)	694 (2.4)	620 (2.4)
Fruit intake	–	–	–	–
None	2344 (10.6)	2029 (8.9)	1701 (6.0)	1045 (4.1)
1 serving per day	8256 (37.5)	8400 (36.7)	9126 (32.1)	6735 (26.3)
2 servings per day	5912 (26.9)	6356 (27.8)	9920 (34.9)	9491 (37.0)
3 or more servings per day	4318 (19.6)	4748 (20.7)	6426 (22.6)	7321 (28.5)
Missing	1182 (5.4)	1360 (5.9)	1232 (4.3)	1056 (4.1)
Psychological distress ^4^	–	–	–	–
Low	8947 (40.6)	11,232 (49.1)	10,275 (36.2)	12,171 (47.5)
Mild	8596 (39.1)	7907 (34.5)	11,499 (40.5)	8564 (33.4)
Moderate	3189 (14.5)	2227 (9.7)	4426 (15.6)	2610 (10.2)
High	856 (3.9)	605 (2.6)	1327 (4.7)	688 (2.7)
Missing	424 (1.9)	922 (4.0)	878 (3.1)	1615 (6.3)
Social interaction ^5^	–	–	–	–
Low	4531 (20.6)	4107 (17.9)	4623 (16.3)	3075 (12.0)
Moderate	12,845 (58.4)	12,912 (56.4)	17,348 (61.1)	14,455 (56.4)
High	3278 (14.9)	4184 (18.3)	4588 (16.2)	6218 (24.2)
Missing	1358 (6.2)	1690 (7.4)	1846 (6.5)	1900 (7.4)
Self-rated health status	–	–	–	–
Excellent	4286 (19.5)	4272 (18.7)	7346 (25.9)	6065 (23.6)
Very good	8862 (40.3)	9733 (42.5)	11,786 (41.5)	11,277 (44.0)
Good	6857 (31.2)	6791 (29.7)	7059 (24.9)	6298 (24.6)
Fair/Poor	1512 (6.9)	1481 (6.5)	1364 (4.8)	1178 (4.6)
Missing	495 (2.2)	616 (2.7)	850 (3.0)	830 (3.2)
Self-rated quality of life	–	–	–	–
Excellent	4229 (19.2)	4201 (18.4)	7229 (25.4)	5945 (23.2)
Very good	8758 (39.8)	9569 (41.8)	11,652 (41.0)	11,070 (43.2)
Good	6755 (30.7)	6625 (28.9)	6952 (24.5)	6149 (24.0)
Fair/Poor	1486 (6.8)	1440 (6.3)	1338 (4.7)	1134 (4.4)
Missing	784 (3.6)	1058 (4.6)	1234 (4.3)	1350 (5.3)
Hypertension	–	–	–	–
No	17,307 (78.6)	15,450 (67.5)	23,902 (84.1)	18,307 (71.4)
Yes	4705 (21.4)	7443 (32.5)	4503 (15.9)	7341 (28.6)
Dyslipidemia	–	–	–	–
No	20,099 (91.3)	19,821 (86.6)	27,107 (95.4)	22,556 (87.9)
Yes	1913 (8.7)	3072 (13.4)	1298 (4.6)	3092 (12.1)
Diabetes	–	–	–	–
No	21,111 (95.9)	21,189 (92.6)	27,572 (97.1)	24,447 (95.3)
Yes	901 (4.1)	1704 (7.4)	833 (2.9)	1201 (4.7)
Arthritis	–	–	–	–
No	21,704 (98.6)	22,278 (97.3)	27,692 (97.5)	24,117 (94.0)
Yes	308 (1.4)	615 (2.7)	713 (2.5)	1531 (6.0)
Asthma	–	–	–	–
No	16,548 (75.2)	17,870 (78.1)	20,900 (73.6)	19,246 (75.0)
Yes	1779 (8.1)	1680 (7.3)	2745 (9.7)	2410 (9.4)

AUD, Australian dollar; TAFE, technical and further education. ^1^ Residential rurality was categorized as four groups including major cities, inner regional area, outer regional area, and remoteness using the Accessibility Remoteness Index of Australia. ^2^ Relative socioeconomic disadvantage was divided into quintiles, with the lowest quintile representing the greatest socio-economic disadvantage. ^3^ Body mass index was calculated as weight in kilograms divided by the square of height in meters. ^4^ Psychological distress measured by the Kessler-10 scale provides a global measure of anxiety and depressive symptoms experienced in the preceding month, with the following categories: low, mild, moderate, and high psychological distress. ^5^ Social interaction was categorized as low, mild, moderate, and high levels using the Duke Social Support Scale.

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
