# Peer review of "Are Leading Risk Factors for Cancer and Mental Disorders Multimorbidity Shared by These Two Individual Conditions in Community-Dwelling Middle-Aged Adults?"

_cancers, 2020, doi:10.3390/cancers12061700_

Round 1

Reviewer 1 Report

This is an interesting article that looks at the risk and risk factors for cancer, mental disorders (depression and anxiety) and concurrent events of these. The use of the 45 and up study allows prospective follow up of a large and linked dataset. The article may benefit from stronger caveats and interpretation.

The effect sizes reported for risk factors and risk of the comorbid conditions relies in part on the response rate to the invitation to participate in the study. This is a seemingly low response rate at baseline. Could the authors comment on the likely impact of this in terms of potential skewed results and implications to their study?

Was there any check with population level data, or population representative data, to see if the low response rate had introduced any sampling bias (decreased the external validity)?

Has there been any validation of using MBS and PBS data to identify incident cancer cases in this cohort, compared with linkage to the NSW cancer registry?

Of interest in this study, is the concurrent presence of cancer and depression and/or anxiety. How have the authors been able to separate the occurrence of depression and/or anxiety as a sequelae of the  cancer diagnosis/treatment and that of depression and/or anxiety that occurs as an additional primary diagnosis (independent of cancer diagnosis and/or treatment)?

The authors are correct in identifying self reported exposures as likely to introduce non-differential misclassification bias into their study. The consequence of this is that significant effects may not be identified. Could the authors consider the impact of such bias on their results?

A surprising finding is the lack of association between physical activity and/or obesity and cancer. These are two well known risk factors particularly for more frequently diagnosed cancers such as breast, colon and prostate cancer. While it has been noted that a meta-analysis has found that for some cancers, there is no association between these lifestyle factors and some cancers, could the authors provide further explanation of their results?

The authors have noted that bowel and prostate cancers were not included in the analysis as there were low numbers of these occurring in the cohort studied. Interestingly, in 2007 prostate and bowel cancer were the most diagnosed cancers in Australia. Are the authors able to provide an explanation why so few cases occurred in their study? What is the likely impact on the interpretation of the results from this study and how does this reflect the generalizability of their results to the broader Australian population? Additionally, if particular cancer sites were not included in the analysis, this should be described in the methods.

Reviewer 2 Report

The present paper aims to identify common determinants for multimorbidity shared by cancer and mental disorders in community-dwelling middle-aged adults. Some major issues need to be addressed before the paper can be published:

  • It needs to be made clear since the beginning that the term multimorbidity is used to refer to the coexistence of cancer and depression. Otherwise one may think it refers to the coexistence of any chronic disease. The title is also confusing for the same reason.
  • In the introduction, the authors state that “determining shared risk factors for mental disorders and cancer would help prevent and manage multimorbidity and minimize deaths caused by these concurrent conditions”. Based on this sentence and the second paragraph of the discussion (and on the use of the term “determinants” in the title and throughout the text), it seems the paper aims to identify causal associations; however, the objective of the study and the applied methods are rather formulated within the prediction framework.
  • Besides cancer and mental health problems, the authors exclude people with heart disease, stroke, dementia or Parkinson’s disease at baseline. Why? They also exclude people aged 65 years or older and those with long-term illness disability. Why? Both cancer and mental disorders are very (and even more) common in old age; thus, excluding people over 65 decreases the generalisability of the findings.
  • Some consideration in relation to the low response rate of the NSW study and its consequences is required.
  • Creating an additional ‘missing’ category grouping missing values for each variable is not recommended as it leads in the best case (if the occurrence of missing values in not related to other variables) to an insufficient adjustment and in the worst case to serious bias in any direction. The authors should consider multiple imputation techniques as appropriate.
  • How was the history, baseline presence and/or incidence of other diseases (beyond heart diseases, stroke, hypertension, diabetes, cancer, Alzheimer’s disease, Parkinson’s disease, depression, arthritis and hip fracture) taken into account? These could be potential confounders of the studied associations, especially if the authors are aiming to infer causality.
  • The selection of potential risk factors needs some justification. Why these and not other factors?
  • The reasoning behind the use of machine learning techniques and the choice of a given set of sensitivity analyses needs to be made explicit in the main text.
  • The lines in Figure 1 cannot be distinguished properly.
  • I do not agree with the following sentence in the conclusion “Individuals with poor/fair self-rated health, and psychological distress may need more care to prevent cancer and mental disorders”. The type of response that subjects with psychological distress require is likely to be found beyond the boundaries of medical care.

Reviewer 3 Report

My major comments are in the modelling part.

The authors have used a few machine learning methods to analyse whether the selected leading determinants for multimorbidity were independently associated with incident cancer and mental disorders. The methodology description is shown n Appendix A. 

Methods such as LR, RF and DL are evaluated in the study. However, since each of those sub-method contains huge variations in the hyper-parameters setting. Using DL as an example, the hyper-parameters could include data normalisation technique, architecture design, optimiser choice, each choice could affect the performance by a large margin. Therefore it is important for the authors to show its parameter setting and the reason behind those design choices behind this study. 

There is no perfect method for modelling, always just more suitable ones. Since are those baselines are not significantly different from each other in terms of performance. I would rather see one method be analysis more deeply than a series of "shallow" trial of various methods not reaching its optimal point. 

Round 2

Reviewer 2 Report

The authors have satisfactorily responded to my comments. I would still suggest the following minor changes:

  • Provide the justification to the exclusion of people with severe diseases, aged 65+ years, and those with long-term illness disability within the methods. The authors already provided an answer, but no changes were carried out in the text.
  • Explicitly mention in the text (perhaps in limitations) that participants in their study were, on average, healthier than the general population in New South Wales.
  • Give further technical details about the multiple imputation in the methods. Additionally, mention that MI of predictors was carried in the titles of figures 2 and 3.
  • Clarify that risk factors were only measured at baseline (this is especially relevant for chronic conditions, which may have developed (i.e. incident comorbidities) throughout the 9-year follow-up).
  • Edit the paper carefully before its publication.

Author Response

The authors have satisfactorily responded to my comments. I would still suggest the following minor changes:

Provide the justification to the exclusion of people with severe diseases, aged 65+ years, and those with long-term illness disability within the methods. The authors already provided an answer, but no changes were carried out in the text.

Response:

We have added the justification in the Methods section:

We excluded people with severe diseases, aged 65 years, or those with long-term illness disability, because they were at high risk of mortality, and these conditions were highly correlated with mental disorders. We excluded those with the Department of Veterans' Affairs cards because information on these people is not included in PBS or MBS.

Explicitly mention in the text (perhaps in limitations) that participants in their study were, on average, healthier than the general population in New South Wales.

Response:

We have mentioned this as a limitation:

Participants in the present study were, on average, healthier than the general population in New South Wales. Similar associations between exposures and health outcomes in this cohort study compared with a population-representative study have been reported.

Give further technical details about the multiple imputation in the methods. Additionally, mention that MI of predictors was carried in the titles of figures 2 and 3.

Response:

We have added more details about the multiple imputation:

Multiple imputations for missing data were conducted for baseline covariates, and we included all variables in the imputation models to create ten imputed datasets.

We have revised the title of Figure 2 as:

Hazard ratios for incident cancer, mental disorders and multimorbidity associated with socioeconomic status, self-rated health and psychological distress, history and family history of chronic conditions after multiple imputation of missing values in covariates.

We have revised the title of Figure 3 as:

Hazard ratios for incident cancer, mental disorders, and multimorbidity associated with behavioral factors after multiple imputation of missing values in covariates.

We have also added an explanation about the comparison between results before and after multiple imputation:

As a minimal difference in HRs (95% CIs) before and after multiple imputation of missing values in covariates was observed, we only present the results after multiple imputation in this figure.

Clarify that risk factors were only measured at baseline (this is especially relevant for chronic conditions, which may have developed (i.e. incident comorbidities) throughout the 9-year follow-up).

Response:

We have clarified this in the Methods section:

All potential predictors included in the prediction model were only measured at baseline.

Reviewer 3 Report

The authors have addressed my concerns about the method part. With further English text proofreading, this manuscript is in an acceptable form. 

Author Response

The authors have addressed my concerns about the method part. With further English text proofreading, this manuscript is in an acceptable form.

Response:

We thank the Reviewer for providing further review, and we have checked out the language throughout the text.